# DNA Methylation Patterns Differ between Free-Living *Rhizobium leguminosarum* RCAM1026 and Bacteroids Formed in Symbiosis with Pea (*Pisum sativum* L.)

**DOI:** 10.3390/microorganisms9122458

**Published:** 2021-11-28

**Authors:** Alexey M. Afonin, Emma S. Gribchenko, Evgeny A. Zorin, Anton S. Sulima, Vladimir A. Zhukov

**Affiliations:** 1Laboratory of Genetics of Plant-Microbe Interactions, All-Russia Research Institute for Agricultural Microbiology (ARRIAM), Podbelsky Sh. 3, 196608 Saint Petersburg, Russia; egribchenko@arriam.ru (E.S.G.); ezorin@arriam.ru (E.A.Z.); asulima@arriam.ru (A.S.S.); vzhukov@arriam.ru (V.A.Z.); 2Sirius University of Science and Technology, Olimpijski Pr. 1, 354340 Sochi, Russia

**Keywords:** *Pisum sativum*, symbiosis, *Rhizobium leguminosarum*, nanopore sequencing, methylation pattern

## Abstract

*Rhizobium leguminosarum* (*Rl*) is a common name for several genospecies of rhizobia able to form nitrogen-fixing nodules on the roots of pea (*Pisum sativum* L.) while undergoing terminal differentiation into a symbiotic form called bacteroids. In this work, we used Oxford Nanopore sequencing to analyze the genome methylation states of the free-living and differentiated forms of the *Rl* strain RCAM1026. The complete genome was assembled; no significant genome rearrangements between the cell forms were observed, but the relative abundances of replicons were different. GANTC, GGCGCC, and GATC methylated motifs were found in the genome, along with genes encoding methyltransferases with matching predicted target motifs. The GGCGCC motif was completely methylated in both states, with two restriction–modification clusters on different replicons enforcing this specific pattern of methylation. Methylation patterns for the GANTC and GATC motifs differed significantly depending on the cell state, which indicates their possible connection to the regulation of symbiotic differentiation. Further investigation into the differences of methylation patterns in the bacterial genomes coupled with gene expression analysis is needed to elucidate the function of bacterial epigenetic regulation in nitrogen-fixing symbiosis.

## 1. Introduction

DNA methylation is an important epigenetic regulatory mechanism in prokaryotes. In addition to protecting the bacterial cell from phages and limiting horizontal transfer by digesting foreign DNA via restriction–modification (R–M) systems [1], methylation plays a significant role in the control of DNA replication and reparation, the cell cycle, and the adaptation due to the so-called phase variations (different methylation patterns in the bacterial population) [2]. For bacteria, DNA methylation is the primary means of epigenetic regulation [3,4]. Methylation systems typically consist of a DNA methylase and one or more DNA binding proteins that can overlap the target methylation site on DNA, subsequently blocking its methylation [5,6,7].

Currently, various aspects of methylation in bacteria are actively studied thanks to the development of technologies for high-precision, real-time sequencing of long molecules [8,9], but there are still large knowledge gaps in this area. For instance, there is very little known about the significance of methylation in such an important process as the endosymbiosis of bacteria and plants.

Soil bacteria from the rhizobia group are able to partake in symbiosis with plants of the Fabaceae family, forming nitrogen-fixing root nodules [10]. Inside these nodules, the bacteria differentiate into the symbiotic form called bacteroids, with the degree of differentiation depending on the plant–microsymbiont pair. In particular, members of the IRL (inverted repeat-lacking) clade of the Papilionoidea subfamily, such as pea, alfalfa, and clover, form the so-called indeterminate nodules, which are considered the most evolutionarily advanced [11,12,13]. Inside these nodules, bacteria undergo terminal differentiation, lose their ability to duplicate, and become metabolically integrated with the host. Bacterial cells grow in size, increase their DNA content, assume irregular shapes, and gain the ability to fix atmospheric N_2_, simultaneously losing the ability to return to a free-living state [14].

Not much is known about bacterial DNA methylation in symbiotic conditions. Although the paper where the changes in adenine methylation were investigated using restriction landmark genome scanning was published in 2006 [15], newer techniques such as third generation (SMRT) sequencing have largely not been used. At the time of writing, only one publication and one preprint were available where the methylation changes in rhizobia within a nitrogen-fixing nodule were investigated using third-generation sequencing. In the article by Davis-Richardson et al., gene expression data were combined with the information about DNA methylation of *Bradyrhizobium diazoefficiens* strain USDA110 in symbiotic nodules of soybean (*Glycine max* (L.) Merr.) [16]; the results showed a specific methylation pattern associated with symbiotic conditions. The preprint by George C. diCenzo et al. describes the methylation patterns in multiple *Ensifer* strains in free-living and symbiotic conditions [17]. Both these works used PacBio SMRT sequencing to investigate the DNA methylation.

In our work, we applied Oxford Nanopore sequencing for genome assembly and methylation analysis of *Rhizobium leguminosarum* (*Rl*) strain RCAM1026 [18], the nitrogen-fixing symbiont of garden pea (*Pisum sativum* L.). This approach allowed us to discern between different types of nucleotides modifications, making it possible to obtain a more detailed picture of DNA methylation.

## 2. Materials and Methods

For the bacteroid isolation, plants of pea cv. Frisson were inoculated with *R. leguminosarum* strain RCAM1026. After vegetation in quartz sand with mineral nutrition [19] for 4 weeks, phenotypically mature (pink) nodules were collected in Eppendorf tubes containing pre-chilled Tris-HCl/sucrose buffer (0.5 M sucrose-50 mM Tris- HCl (pH 8.0), dithiothreitol (10 mM), and polyvinylpolypyrrolidone (5%)). The nodules were ground using mortar and pestle; the entire bacteroid isolation procedure, with the exception of the final stage, was carried out in sucrose buffer, as in [20]. To remove the plant cells debris, the crushed nodule material was filtered through a Miracloth-like material, followed by washing in the same buffer. The resulting suspension was centrifuged for 1 min at 10,000× *g* at +4 °C. The symbiosome-containing pellet was resuspended in Tris-HCl/sucrose buffer. The resulting suspension was distributed into several Eppendorf tubes containing a “sucrose cushion” consisting of 1.5 M sucrose and 50 mM Tris-HCl (pH 8.0) and centrifuged for 30 s at 5000× *g*, +4 °C. The upper phases enriched with symbiosomes were transferred into one polypropylene tube and centrifuged for 90 s at 10,000× *g*, +4 °C, which made it possible to concentrate the symbiosome fraction into a pellet. The pellet was then resuspended in Tris-HCl/sucrose buffer and again applied to the “sucrose cushion” followed by centrifugation for 5 min at 10,000× *g*, resulting in precipitation symbiosomes with bacteroids. To remove the peribacteroid membrane and isolate the bacteroids, the pellet was resuspended and centrifuged in a buffer containing Tris-HCl, but without sucrose. The resulting precipitate was frozen in liquid nitrogen and stored at −80 °C for subsequent DNA extraction.

For DNA isolation from the cell culture, one colony of *Rl* RCAM1026 was grown on an orbital shaker in a 50 mL flask with 10 mL of TY media for 18 h at 28 °C, 200 rpm. The cell density, as measured on the NanoDrop OneC spectrophotometer (Thermo FS, Waltham, MA, USA), was 0.4. The cell culture and the bacteroids from the cv. Frisson nodules (see above) were used for DNA isolation.

DNA from bacteroid samples and from the cell culture was isolated using a modified phenol-chloroform method as described in [21].

To obtain non-methylated DNA, the following whole-genome DNA amplification procedure was performed. The DNA isolated from the cell culture was sheared using a 32-gauge Meso-Relle^®^ hypodermic needle (Biotekne S.R.L., Bologna, Italy). Fragment length analysis by the TapeStation system 4150 (Agilent, Santa Clara, CA, USA), software revision 3.1.1, showed the fragment distribution to be centered at around 13000 b.p. A total of 30 ng of the sheared DNA was used for end-prep reaction using the NEBNext^®^ Ultra™ II End Repair/dA-Tailing (E7546, New England Biolabs, Ipswich, MA, USA). A double strand DNA fragment was formed by heating two oligonucleotides (NP_adapt_2_fw 5′ AAAGACAACCACGACTATAACGT 3′ and NP_adapt_2_rv 5′ CGTTATAGTCGTGGTTGTCTTT 3′) to 95 °C in a water bath and letting them cool down at room temperature. The fragment was ligated to the end-prepped DNA using the Blunt/TA ligase (M0367, New England Biolabs, Ipswich, MA, USA) according to the manufacturer’s instructions. The NP_adapt_2_fw primer was used to amplify the DNA for sequencing. For the PCR reaction LongAmp (M0287, New England Biolabs, Ipswich, MA, USA) Taq mix was used, with the following reaction parameters: initial denaturation 94 °C for 60 s, then 8 cycles of 94 °C for 30 s, 58 °C for 8 min, and final extension of 60–65 °C for 10 min. The PCR amplification was sufficient to obtain 150 ng of WGA DNA. The PCR reaction product was cleaned using AMPureXP beads (Beckman Coulter, Brea, CA, USA) and the resulting DNA was used for sequencing. The resulting DNA fragments were considered non-methylated, and the obtained data were used as non-methylated controls in subsequent analysis after additional filtering.

Long-read whole genome sequencing was performed using a MinION sequencer (Oxford Nanopore Technologies, Oxford, UK) in the Core Center “Genomic Technologies, Proteomics and Cell Biology” in All-Russia Research Institute for Agricultural Microbiology (ARRIAM). The SQK-LSK109 Ligation Sequencing Kit with the EXP-NBD104 Native Barcoding Expansion 1-12 kit (Oxford Nanopore Technologies, Oxford, UK) were used to prepare the library according to manufacturer’s instructions, omitting the DNA shearing step.

The reads were base-called and demultiplexed using the Guppy_basecaller (v.5.0.5). The genomes were from the barcoded reads with Flye (v.2.9) [22] and polished with Racon (v.1.3) [23], Medaka (v.1.4.3), and Pilon (v.1.23) [24], as in [25]. The WGA reads were checked for the NP_adapt_2_fw barcode sequence on both ends, the sequence was removed, and these reads were used for the analysis.

The genome comparison was performed using MUMMER (v.4.0.0) [26]. Sniffles (v.1.0.12a) and cuteSV (v.1.0.12) structural variation callers were used for the finding of genome rearrangements [27,28], with minimap (v.2.17) used for long read mapping [29].

Megalodon toolkit (v.2.3.4) was used for methylation calling and pattern finding in the genome. The res_dna_r941_min_modbases-all-context_v001.cfg configuration file from the rerio (https://github.com/nanoporetech/rerio, accessed on 10 August 2021) was used to obtain the methylation status of adenine and cytosine. The data were then visualized using custom R (v.4.1.1) scripts. Additional methylation calling and motif analysis was performed using the Nanodisco (v.1.0.3) pipeline [30]. The genome annotation was performed using PGAP (v.2021-07-01.build5508) [31], and functional COG [32] annotation was performed using eggNOG-mapper [33], based on eggNOG orthology data [34]. Sequence searches were performed using [35]. Enrichment analysis was performed in the clusterProfiler, and fdr was used for calculating the adjusted p-value [36]. The restriction enzyme database REBASE [37,38] was used to search for potential methyltransferase genes in the genomes of *Rl* strain RCAM0126.

## 3. Results

### 3.1. Genome Assembly and Comparison

In this study, the genome of the Rl strain RCAM1026 was assembled de novo using the long reads. Nanopore sequencing yielded 283 m.b.p. of reads with N50 = 30,784 for the cell culture, and 1.4 g.b.p with N50 = 5756 for the bacteroids; the reads were used for genome assembly as described in Section 2. The genomes were compared using the MUMMER dnadiff script. No large genome rearrangements between the genome assemblies of the bacteroids and the cell culture were found, and the lack of rearrangements was additionally confirmed using the Sniffles algorithm. The cuteSV analysis showed a number of structural variations. Most variations were the same for the two analyzed conditions. Large deletions were observed in both conditions on the chromosome at 1,758,497 (763 b.p.) and 2,470,634 (2081 b.p.). Insertions were observed on the chromosome at 2,624,645 (80 b.p.), on pRL10 at 172,084 (48 b.p.), on pRL11 at 344,926 (53 b.p.), and on pSym at 46,522 (65 b.p.). Two structural variations were observed only in the bacteroids: a duplication on the chromosome at 3,132,085 (2045 b.p.), and a deletion on pRL12 at 311,804 (1185 b.p.). No structural variations were exclusive to the free-living state. The duplication occurred in a genome location annotated as containing multiple VCBS containing proteins. The deletion on the pRL12 covered parts of the two genes, BSO17_31670 and BSO17_31675, and formed a protein 259 a.a. long without known homologues in the NCBI database. This result, however, was reported only by cuteSV and should be verified be external evidence.

The RCAM1026 genome was assembled into 5 circular replicons. The sequences were rotated so that the chromosome started with the Ori sequence, and the other replicons began with the RepA gene. The resulting assembly consisted of one chromosome and 4 plasmids (Table 1).

Although there were few structural differences between the free-living and symbiotic genomes, there was a significant difference in coverage. Since the libraries were prepared using native DNA, the coverage should correspond to the number of copies of the replicon. The coverage diagram for the cell culture and the bacteroid conditions (Figure 1) show uniform coverage of all the replicons. Relative number of replicon copies in cells, however, differed between the conditions (Table 1), with the symbiotic plasmid (carrying the nod and nif clusters) being under-represented in the cell culture and the chromosome being over-represented in the bacteroids.

### 3.2. Methylation Motif Search

The cell culture genome assembly was used for this and subsequent steps. Methylation motives in the analyzed samples were searched for using Nanodisco pipeline. GANTC, GATC, and GGCGCC methylated motifs were found to have a modified base, as previously described in Afonin et al. (2021; thesis, under review). [39] The precise mapping results are shown in Table 2. The GANTC motif showed a strong signal for the 6mA base, which was in concordance with our expectation, as this type of methylation has been previously reported for α-proteobacteria [2]; however, methylation of the GGCGCC motif has been not reported for the *Rhizobuim leguminosarum* species complex.

Since methylation can play a role in gene regulation, it was important to investigate whether the detected motifs are preferentially present in the promoter regions of the RCAM1026 genes. The results of the analysis for all three motifs are presented in Table 3.

Although the GANTC and the GGCGCC motifs are present in very similar numbers in the genomes, the GANTC motif is much more frequently found in the promoter regions, especially on the plasmids. Genes with this motif in their promoter are likely to be regulated with GANTC methylation and play a role in the cell cycle. To test this, an enrichment test of these genes according to COG categories was performed. Only GANTC containing genes were significantly enriched in the L category (replication, recombination, and repair), padj = 0.02.

### 3.3. Methylation System Genes in the RCAM1026 Genome

The annotated proteins in the RCAM1026 genome were BLAST-searched against the REBASE database. The gene most likely responsible for the methylation of GANTC motif is RCAM1026_000980 (97% similarity to *M.retCII*). We found two copies of a methylase with predicted target motif GGCGCC (~52% similarity to *M.CcrNAIV*)—RCAM1026_001000 on the chromosome and RCAM1026_005751 on the pRL11 plasmid. In both locations, the GGCGCC methylase gene was situated next to a Putative Type II restriction enzyme gene, suggesting the usage of the GGCGCC motif as a target for a restriction–modification system. Two copies of methyltransferase gene with predicted target motif GATC were found on the chromosome. The predicted modifications according to the database are G6mATC for RCAM1026_002430, and GAT5mC for RCAM1026_000801. All the found genes involved in methylation and their possible motifs are presented in Table 4. All the found genes were classified as belonging to Type II DNA methylation systems.

### 3.4. Genome-Wide Methylation Patterns

The precise methylation levels for each A (adenine) and C (cytosine) base as reported by the Megalodon pipeline were used for analysis of the DNA methylation. A model tuned for discerning A and C methylation simultaneously was used for methylation calling. As each read produced by the MinION sequencer corresponds directly to a fragment of native DNA, the data were not normalized. The levels of nucleotide methylation across the genome are presented in Figure 2.

The analysis shows that the levels of A methylation in the symbiotic state were much higher than in the free-living state, while the C methylation was much more similar between the two states.

The methylation patterns of the GGCGCC motif in two conditions are presented in Figure 3. This motif was mostly methylated in all the replicons in both conditions, and average cytosine methylation in this motif was observed to be at 98%, compared to averages of 46% in bacteroids and 38% in cell culture.

The methylation patterns of the GANTC motif in two conditions are presented in Figure 4. In the bacteroids, the adenine methylation of the GANTC motif was at around 98% on all the chromosomes, higher than the 75% average for the A methylation, concordant with preferential methylation of A in this motif. For the cell culture, the GANTC motif showed increased methylation on the Ter region of the chromosome (the middle of the chromosome); no such pattern was observed for other replicons.

The methylation patterns of the GATC motif in the two conditions are presented in Figure 5. The methylation patterns for this motif were very different from those of the GANTC and GGCGCC motifs in both the investigated conditions. The methylation for this motif in bacteroids was around 50% for the chromosome and 40% for the plasmids. For the cell culture, the GATC motif was the only one with lower methylation than on average for the respective base.

## 4. Discussion

Using Oxford Nanopore sequencing technology, the genome of the *Rhizobium leguminosarum* strain RCAM1026 was investigated in two different states—free-living cells, and terminally differentiated bacteroids. The analysis did not show any large-scale genome rearrangements (in-dels, transpositions, or loss of replicons) taking place during terminal differentiation of cells. However, the genome coverage analysis showed significant differences in the relative abundances of replicons between the investigated states. The lower observed copy number of the symbiotic plasmid in free-living culture illustrates the relative ease with which the strain loses its symbiotic plasmid when propagated on solid media without the signals from a possible host for a prolonged period of time. The observed phenotype may be a naturally occurring population structure in which only a part of the population carries the plasmid, making the whole population more resistant to the changing environmental conditions [40].

The equalization of the plasmid copy number in the bacteroids is to be expected, as only the cells possessing the full array of symbiotically critical genes should be able to enter the symbiotic state. The relatively higher coverage of the chromosome observed is similar to the observation made by George C. diCenzo et al. in [17], where a similar pattern was described for the replicons in the bacteroids formed by *Ensifer* bacteria. Although the observed coverage should be in direct relation with the chromosome copy number in free-living cells and bacteroids, the differences in DNA content in the bacteroids need to be verified directly in further studies. The exact reason behind this abundance variation is unclear; one possible explanation is that the preferential replication of the chromosome makes it possible for the bacteria to increase the transcription of important metabolic genes by increasing the number of gene copies in the cell.

Three motifs with significant methylation percentage were revealed: GANTC, GGCGCC, and GATC. For all the motifs, corresponding genes encoding methyltransferase enzymes were found in the genome. A few additional methyltransferase genes were also found, which is consistent with the overall methylation patterns for the adenine and cytosine not coinciding completely with the methylation patterns of the found motifs.

The Nanodisco pattern characterization showed GANTC to contain a 6 mA base in the second position; GGCGCC and GATC contain 4 mC bases at the third and fourth position, respectively. The methylation patterns in the two studied conditions are quite different. The 6 mA is much more likely to occur in the bacteroids than in the cell culture, leading us to believe that 6 mA is somehow involved in the terminal cell differentiation. Adenine-specific methylation has been well studied and shown to have diverse cellular roles [41,42]. Indeed, the GANTC motif, known to be involved in the cell cycle regulation in proteobacteria [2], is fully methylated in the bacteroid genome, which is consistent with the termination of cell division in the bacteroids. The genes, containing GANTC motifs in their promoter regions, were significantly enriched in the “replication, recombination, and repair” COG category, which is also to expected. The bow-like pattern of GANTC methylation observed in the chromosome is in concordance with the fact that the methylation of DNA is dependent on the motif position in respect to replication origin (Ori site). No such pattern was observed for the plasmids.

The pattern of cytosine methylation is much more similar in the two conditions. Cytosine methylation seems to be less connected to the cell cycle and differentiation status. However, two found motifs with methylated cytosine behave very differently.

The GGCGCC pattern is most likely connected to the restriction–modification system (R–M). Taking into account the two annotated restriction–modification clusters, one on the chromosome and the other on the pRl11 plasmid, the system highlights the importance of this pattern for *R. leguminosarum*. The presence of the R–M system on the pRL11 plasmid may serve as a mechanism of plasmid persistence in bacterial cells [43]. However, since the chromosome also has the R–M system with the same target motif, this mechanism would not work in this case. The plasmid R–M system may work in tandem with the chromosomal one, both participating in regulatory processes in the bacterial cell, possibly by enforcing a specific pattern of methylation [44].

Compared to the other two motifs, the GATC motif shows a unique pattern of methylation. The levels of methylation in cell culture are lower for this motif than for the cytosine on average, while in the bacteroids these levels are higher. This points at the possible activation of GATC-recognizing methyltransferases in bacteroids, making GATC motif a prospective candidate for the terminal differentiation regulator.

## 5. Conclusions

Rapid development of single-molecule sequencing technologies opens up the prospects of deeper understanding of the genetic and epigenetic mechanisms that govern the living cells. In this work, we demonstrate that the differentiated bacteroids within the cells of pea symbiotic nodules have different methylation profiles compared to free-living bacterial culture, but do not have any DNA rearrangements such as deletions and/or duplications of parts of chromosomes or plasmids. On the other hand, the overall coverage of replicons was not uniform, which possibly reflects the loss of the Sym plasmid in part of the bacterial population in free living culture, and higher levels of chromosome replication in bacteroids. The observed coverage differences for the symbiotic plasmid, previously also reported for the *Rl* strain A1 [25], may speak in favor of the possible heterogeneity of some *Rl* populations.

Among the detected DNA methylation motifs, we found those present in both bacteroids and free-living cells. The discovery of corresponding DNA methyltransferase genes most likely targeting all found motifs in the RCAM 1026 genome indirectly supports our methylation pattern search and analysis. The GANTC motif was found in a large proportion of the gene promoters, especially for genes linked to cell cycle control, and was methylated in both the cell states, showing that *Rl* is not an exception from the previously described pattern. The existence of two restriction–modification system protein groups with target motif GGCGCC point at the importance of this system for the cell. The mechanism of the functioning of this R–M system is unclear.

The GATC motif is the least studied of the three motifs found. Its exact function is unclear, but the preferential methylation of this motif in the bacteroid cells and very low methylation percentage in free-living conditions suggest some form of state-dependent activation mechanism of the found GATC methylase.

Since DNA methylation serves as a mechanism for regulation of gene expression, we expect that combining our data with comprehensive transcriptome analysis will definitely help understand the genetic and epigenetic bases of bacteroid differentiation in the cells of legume nodules.

## Figures and Tables

**Figure 1 microorganisms-09-02458-f001:**
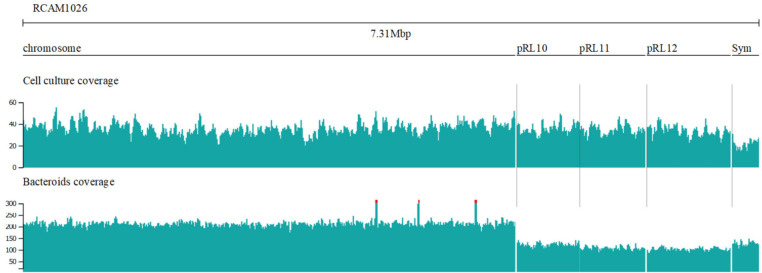
The coverage plots for the genome in two conditions. The plots represent the coverage statistics across the replicons.

**Figure 2 microorganisms-09-02458-f002:**
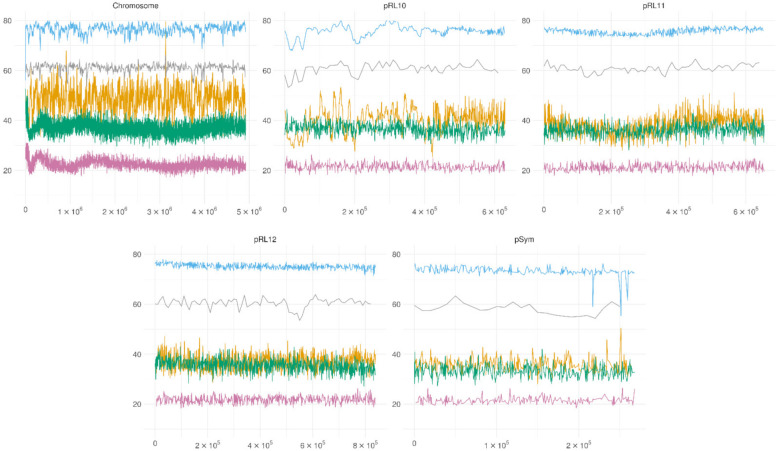
DNA methylation patterns. The plots represent the percentage of methylated adenine or cytosine in the *R. leguminosarum* RCAM1026 genome shown across a 5 kb sliding window. The y scale shows the percentage of methylated reads mapped to the position. Blue—mA in bacteroids, purple—mA in cell culture, yellow—mC in bacteroids, green—mC in cell culture, grey—GC content.

**Figure 3 microorganisms-09-02458-f003:**
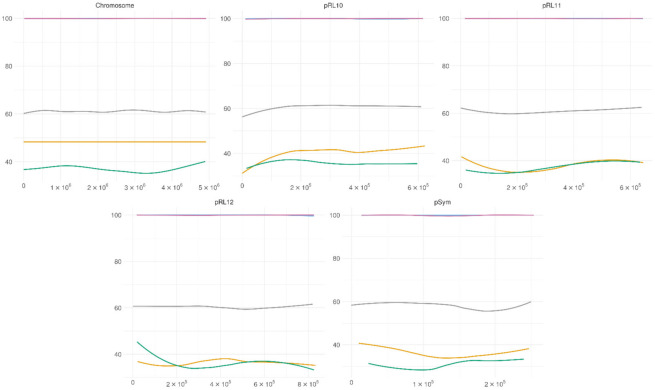
GGCGCC methylation across the replicons. Blue—GGCGCC motif methylation in bacteroids, purple—GGCGCC motif methylation in cell culture, yellow—methylation of cytosine in bacteroids, green—methylation of cytosine in bacteroids, grey—GC content. Solid lines—polynomial regression lines calculated using ggplot_smooth “loess” method, formula “y~x”.

**Figure 4 microorganisms-09-02458-f004:**
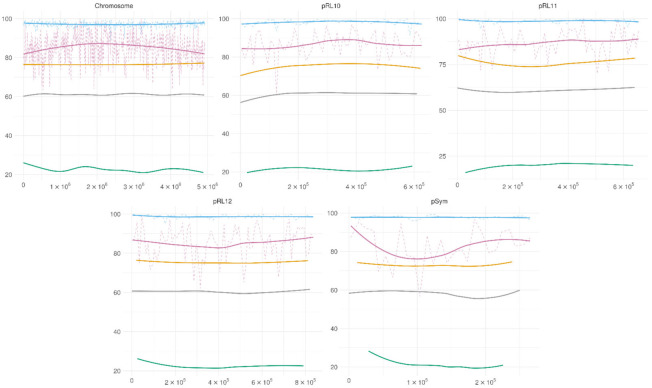
GANTC methylation across the replicons. Blue—GANTC motif methylation in bacteroids, purple—GANTC motif methylation in cell culture, yellow—methylation of adenine in bacteroids, green—methylation of adenine in cell culture, grey—GC content. Solid lines—polynomial regression lines calculated using ggplot_smooth “loess” method, formula “y~x,”; dotted lines represent the average methylation across a 5 kb sliding window.

**Figure 5 microorganisms-09-02458-f005:**
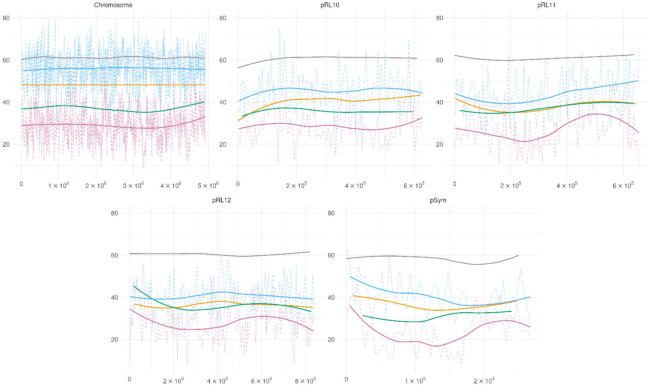
GATC methylation across the replicons. Green—GATC methylation in bacteroids, violet—methylation in cell culture, purple—methylation of cytosine in bacteroids, blue—methylation of cytosine in bacteroids, grey—GC content. Solid lines—polynomial regression lines calculated using the ggplot_smooth “loess” method, formula “y~x”, dotted lines—the average methylation across a 5 kb sliding window.

**Table 1 microorganisms-09-02458-t001:** Statistics for the genome assembly and annotation of the strain RCAM1206.

Chromosome Name	Length	CDS	tRNA Genes	GC Composition	Coverage in Bacteroids	Coverage in Cell Culture ^1^
Chromosome	4,921,456	4736	51	61.09	1	1
pRL10	629,474	575	2	60.61	0.91	0.58
pRL11	655,637	625	-	60.89	0.94	0.5
pRL12	838,366	756	-	60.40	0.97	0.48
Symbiotic plasmid	268,924	263	-	58.12	0.59	0.6

^1^ The replicon coverage was calculated in relation to the chromosome for each condition.

**Table 2 microorganisms-09-02458-t002:** Motif distribution by replicon.

Motif	Characterized Motif	Modified Base	Modified Position	Nanodisco Score
GANTC	G6mANTC	6mA	2	34.16
GATC	GAT4mC	4mC	4	35.68
GGCGCC	GG4mCGCC	4mC	3	41.49

**Table 3 microorganisms-09-02458-t003:** Methylation patterns intersection with genes and gene promoter regions.

Motif	Chromosome	pRL10	pRL11	pRL12	pSym
GANTC	6873 (841) ^1^	770 (96)	778 (136)	1037 (149)	406 (65)
GATC	45,690 (6029)	5963 (633)	6042 (595)	7889 (767)	2264 (252)
GGCGCC	6466 (812)	756 (45)	754 (57)	888 (81)	213 (23)

^1^ The number of motifs found in promoters is indicated in parentheses. Gene promoter regions were defined as 100 b.p. upstream of the start codon of each gene.

**Table 4 microorganisms-09-02458-t004:** Genome methylation systems in RCAM1026.

Gene	Replicon	Putative Enzyme Type	Motif	Homologue in REBASE	Similarity
000897	Chromosome	Methyltransferase	-	M.MspCH12ORF7910P	54.422
000974	Chromosome	Methyltransferase	-	M.Hhe1ORF5290P	59.823
000982	Chromosome	Methyltransferase	GATC	M.MspME121ORFAP	51.515
001161	Chromosome	Methyltransferase	GANTC	M.RleNORF744P	100.000
001164	Chromosome	Nicking endonuclease	-	N.Pec32ORF2247P	56.267
001182	Chromosome	Methyltransferase	GGCGCC	M.CspK31ORF2261P	70.984
001183	Chromosome	Restriction enzyme	-	Avi39ORF4780P	64.912
001184	Chromosome	Restriction enzyme	-	Sma240ORF2946P	75.431
001185	Chromosome	Nicking endonuclease	-	V.OspA1ORF4070P	69.919
001185	Chromosome	Helicase domain protein	-	H.AspSLV7ORF8235P	82.676
001515	Chromosome	Methyltransferase	-	M.CspCJ34ORFGP	50.882
001573	Chromosome	Methyltransferase	-	M.Hhe1ORF5290P	58.850
002613	Chromosome	Orphan methyltransferase	GATC	M.Sen6759Dam	50.935
002688	Chromosome	Methyltransferase	-	M.EcoF3113ORF24645P	57.277
003957	Chromosome	Methyltransferase	-	M.EcoF3113ORF24645P	68.584
005751	pRl11	Restriction enzyme	GGCGCC	M.SfrNXT3ORF1642P	89.922
005752	pRl11	Methyltransferase	GGCGCC	SfrNXT3ORF1642P	92.014
005755	pRl11	Nicking endonuclease	-	V.OspA1ORF4070P	67.424
006861	pRl12	Methyltransferase	-	M.Hhe1ORF5290P	60.526

## Data Availability

This Whole Genome Shotgun project has been deposited at GenBank under the accession CP084696-CP084700. The raw Illumina data are deposited under the accession SRR16242825, Nanopore data (raw signal data) for the cell culture—SRR16229315 (SRR16229311), WGA—SRR16229312 (SRR16229309), bacteroids SRR16229313 (SRR16229310). The version used in this paper is the second version of the genome.

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
