# Peer review of "DNA Methylation Patterns Differ between Free-Living Rhizobium leguminosarum RCAM1026 and Bacteroids Formed in Symbiosis with Pea (Pisum sativum L.)"

_microorganisms, 2021, doi:10.3390/microorganisms9122458_

Round 1
Reviewer 1 Report
The manuscript is well written and flows well. Bioinformatics methods used are up to date and competently performed with Megalodon, nanodisco etc, and this will get the best out of the data. This is good to see.
One concern would be the limited biological relevance of the (nice) data. Correlation of methylation states and genomic features / genes would greatly enhance the importance of the manuscript.
I note however relatively low sequencing yields, 0.2 and 1.5 GB. Why are yields so unexpectedly low ? I have seen 20+GB per 9.4.1 flowcell for bacteria.
Was there a problem preparing HMW DNA ?
Coverage seen here (~30 - 200 or so in the images) is fine for methylation and SV analyses however.
Sniffles is a widely used SV caller, but has some weaknesses, especially with bacterial genomes in our experience.
Alternative callers include eg cuteSV used in ONTs best practice https://github.com/nanoporetech/pipeline-structural-variation .
Results:
"
Although there were no structural differences between the free-living and symbiotic 157
genomes, there was a difference in coverage. The coverage diagram for the cell culture 158
and the bacteroid conditions (Figure 1) show uniform coverage of all the replicons. Rela- 159
tive coverage, however, differed between the conditions (Table 1), with the symbiotic plas- 160
mid (carrying the nod and nif clusters) being under-represented in the cell culture and the 161
chromosome being over-represented in the bacteroids.
"
I think this coverage difference should be referred to as a potential difference of copy-number of the non-chromosomal replicons. Analously to CNVs in cancer genome analysis.
I would suggest rephrasing the discussion of coverage and talk more about copy-numbers, which is much more understandable.
Also the very nice coverage graphs here make the coverage discussion far clearer.
I find Fig2 impossible to read (in the version I have) so cannot really evaluate.
Methylation analysis. I found these parts too global and would like to see more local analyses of genes, promoters and other features if available.
Which genes are most likely to be affected by these methylated marks ? Are any examples useful for physiological interpretation ?
Discussion:
"
However, the genome coverage analysis showed significant differ- 257
ences in the relative abundances of replicons between the investigated states. The lower 258
coverage of the symbiotic plasmid in free-living culture illustrates the relative ease with 259
which the strain loses its symbiotic properties when propagated on solid media for a pro- 260
longed period of time.
"
Again, even if this is true - and it may be, the coverage is a primarily a technical issue. Lower coverage may indicate biological importance and lower
cell counts (a al metagenomics) but only if technical issues are first removed.
For example, the authors could stress the point that
a) coverage differences were not due to different flowcells used or other technical issues
b) coverage per chromosome shows similar patterns, yet only coverage and therefore putative copy number differences of the replicons are implicated
Perhaps also an idea - do the replicons have single copy genes which show "heterozygous" phased SNVs in part indicating that several genetically distinct clones are present ?
Just speculation, but could back up the authors copy number hypothesis.
Conclusions:
"
The observed coverage differences for the symbiotic plasmid, previ- 320
ously also reported for the Rl strain A1 [25], may speak in favor of the possible heteroge- 321
neity of bacterial population in nature being somehow beneficial.
"
This is highly speculative and completely lacks evidence.
Author Response
"Please see the attachment.

Reviewer 2 Report
This is well-rounded manuscript about one particular problem regarding possible methylation function in rhizobia legume symbiosis. Language is concise and easy-to-read. I recommend to publish it in it's present state, but I have a couple of remarks for the authors.
- The reason for using the particular Rl RC1AM026 strain is unclear
- Did you sterilize nodules before bacteroids isolation? There should be plenty of undifferentiated cells outside of the nodule
- It seems that manuscript focuses mostly on methodological problems, and not biological. Although the article answers all the questions it puts in the beginning, the question itself is not entirely explains its biological significance.
AK
Author Response
This is well-rounded manuscript about one particular problem regarding possible methylation function in rhizobia legume symbiosis. Language is concise and easy-to-read. I recommend to publish it in it's present state, but I have a couple of remarks for the authors.
Thank you for reviewing the manucript!
- The reason for using the particular Rl RC1AM026 strain is unclear
RCAM1026 is used as a growth promoter and as an inoculant in the experiments performed in ARRIAM. Although it is both not the type strain, and not the most widely accepted laboratory strain, it a) belongs to the biggest R.leguminosarum genospecies (C), b) is capable of fixing nitrogen highly efficiently, while also possessing some unique phenotypes when in symibiosis with mutant pea lines, and c) was sequenced by our group, making it easier for us to work with.
- Did you sterilize nodules before bacteroids isolation? There should be plenty of undifferentiated cells outside of the nodule
Since the nodules contain non-differentiated bacteria, we thought it impossible to get rid of all the bacteria present in, for example, infection threads and infection zone. We relied mainly on the nodule isolation procedure, which uses sucrose gradient to leave us with mostly bacteroids. For further investigation we will possibly split the nodules into zones and isolate the bacteroids separately from each zone, thus further minimizing possible contaminations.
- It seems that manuscript focuses mostly on methodological problems, and not biological. Although the article answers all the questions it puts in the beginning, the question itself is not entirely explains its biological significance.
The article answers two main biological questions, not previously investigated -
1) are there changes in the genomes and the epigenomes of the bacteria undergoing terminal differentiation
2) what are the patterns and motifs of methylation in the bacterial genomes, and do those change during the terminal differentiation.
We discovered the differences in the copy number variations in the genome of the investigated strain dependent on the differentiation state. We discovered methylation motifs as well as corresponding methylation system genes. These are the biologically significant findings. We will try to continue the investigation of this particular symbiosis using transcriptomics and more detailed methylation and differential methylation analysis.
